# Transgenic Sweet Orange Expressing the *Sarcotoxin IA* Gene Produces High-Quality Fruit and Shows Tolerance to ‘*Candidatus* Liberibacter asiaticus’

**DOI:** 10.3390/ijms23169300

**Published:** 2022-08-18

**Authors:** Talita Vigo Longhi, Deived Uilian de Carvalho, Izabela Moura Duin, Maria Aparecida da Cruz, Rui Pereira Leite Junior

**Affiliations:** 1Área de Proteção de Plantas, Instituto de Desenvolvimento Rural do Paraná—IAPAR/Emater (IDR-Paraná), Celso Garcia Cid Road, km 375, Londrina 86047-902, PR, Brazil; 2Centro de Ciências Agrárias, Universidade Estadual de Londrina (UEL), Celso Garcia Cid Road, km 380, Londrina 86057-970, PR, Brazil; 3Departamento de Pesquisa e Desenvolvimento, Fundo de Defesa da Citricultura (Fundecitrus), 201 Dr. Adhemar Pereira de Barros, Araraquara 14807-040, SP, Brazil

**Keywords:** *Citrus* × *sinensis* (L.) Osbeck, biotechnology, genetically modified citrus (GMC), antimicrobial peptide (AMP), huanglongbing (HLB), fruit quality

## Abstract

Huanglongbing (otherwise known as HLB or greening) is currently the most devastating citrus disease worldwide. HLB is primarily associated with the phloem-inhabiting bacterium ‘*Candidatus* Liberibacter asiaticus’ (*C*Las). Currently, there are no citrus species resistant to *C*Las. Genetic transformation is one of the most effective approaches used to induce resistance against plant diseases. Antimicrobial peptides (AMPs) have shown potential breakthroughs to improve resistance to bacterial diseases in plants. In this paper, we confirm the *Agrobacterium*-mediated transformation of Pera sweet orange expressing the AMP *sarcotoxin IA* (*stx IA*) gene isolated from the flesh fly *Sarcophaga peregrina* and its reaction to *C*Las, involving plant performance and fruit quality assessments. Four independent transgenic lines, STX-5, STX-11, STX-12, and STX-13, and a non-transgenic control, were graft-inoculated with *C*Las. Based on our findings, none of the transgenic plants were immune to *C*Las. However, the STX-5 and STX-11 lines showed reduced susceptibility to HLB with mild disease symptoms and low incidence of plants with the presence of *C*Las. Fruit and juice quality were not affected by the genetic transformation. Further, no residues of the sarcotoxin IA protein were found in the juice of the STX-11 and STX-12 fruits, though detected in the juice of the STX-5 and STX-13 lines, as revealed by the immunoblotting test. However, juices from all transgenic lines showed low traces of sarcotoxin IA peptide in its composition. The accumulation of this peptide did not cause any deleterious effects on plants or in fruit/juice. Our findings reinforce the challenges of identifying novel approaches to managing HLB.

## 1. Introduction

Brazil is the world’s largest producer of sweet orange (*Citrus*
*× sinensis* (L.) Osbeck) fruit and juice, accounting for three-quarters of the total sweet orange juice exports [1,2]. Historically, the Brazilian citrus industry has faced challenges due to intense disease pressure favored by the humid subtropical and tropical climates [3,4,5]. Nowadays, huanglongbing (known as HLB or greening) is considered the most devastating and widespread citrus disease worldwide, as no cure or citrus resistance has been reported to date [4,5,6,7]. This disease is present in Asia, America, and Africa, affecting all citrus species, cultivars, hybrids, and relatives [4]. In Brazil, HLB was first reported in 2004 [8].

Three species of Gram-negative phloem-limited α-proteobacteria ‘*Candidatus* Liberibacter’ are associated with this disease, ‘*Ca.* Liberibacter asiaticus’ (*C*Las), ‘*Ca.* Liberibacter americanus´ (*C*Lam), and ‘*Ca.* Liberibacter africanus´ (*C*Laf) [9,10], which affect all citrus species [4,11,12]. The Asian species *C*Las is the most aggressive and overspread through the main citrus-growing areas around the world [4]. This bacterium is transmitted by the Asian citrus psyllid (ACP) *Diaphorina citri* Kuwayama (Hemiptera: Liviidae), as well as by the grafting of infected plant material [13,14]. ‘*Ca*. Liberibacter spp.’ colonize the phloem vessels and modulate different physiological processes and pathways in citrus plants, including photosynthesis, the transport of nutrients, hormone balance, carbohydrate metabolism, plant–water relations, and others [7,15,16,17].

The development of HLB symptoms in citrus plants is triggered by a multilayered metabolic network, which is governed by phytohormones [7]. Symptoms may vary according to the citrus genotype, age and physiological stage of the plant, inoculation pressure, and season of the year [18,19]. The leaves of HLB-affected plants usually show a blotchy mottled chlorotic pattern of thicker and leathery aspect with enlarged corky yellow veins, asymmetric and reduced in size [4,20]. HLB-affected fruit is usually smaller, lopsided with a curved axis, and has small brownish aborted seeds [11,17,20,21,22]. Moreover, HLB-affected fruit tends to drop prematurely, exhibiting a color inversion and an imbalance in carbohydrate and hormone metabolism [7,17]. Juice from HLB-affected fruit is generally higher in acid content, lower in soluble solids and sugar–acid ratio, and has several off-flavors [11,23,24,25,26,27,28].

A major concern regarding HLB is related to the limited resources for its management, which involves the frequent spraying of different chemicals for controlling the insect vector ACP, and eradication of symptomatic plants [29]. Indeed, this pathosystem may lead the citrus industry to collapse, as the affected plants lose vigor quickly and show a drastic reduction in yield. Further, plants decline and die a few years following bacterial infection [8,26,30]. Under this context, the development of better horticultural practices is of paramount importance to tackle the negative impact of HLB on citriculture.

Genetic transformation is an alternative to obtaining “genetically modified citrus” (GMC) with a certain level of resistance to HLB [5]. GMC plants may carry foreign genes that could interfere with bacterial growth and/or in psyllid activity [5]. Under this context, the insertion of specific genes that inhibit virulence factors in the pathogen, enhance natural defense mechanisms in the host, and/or encode antimicrobial peptides (AMPs) may have some potential to develop GMCs with resistance against bacterial diseases [31,32,33,34,35,36,37].

Among the antimicrobial peptides (AMPs), sarcotoxin IA (STX IA) is a peptide secreted by the larvae of the flesh fly *Sarcophaga peregrina* in response to bacterial infection [38]. This peptide isolated from the hemolymph of the flesh fly [39,40,41] has shown effective antibacterial properties in plants, including the bacterium *Xanthomonas citri* subsp. *citri*, the citrus canker causal agent [37,39]. The sarcatoxin IA consists of 39 amino acid residues and two amphiphilic α-helical regions, i.e., helix I (Leu^3^–Gln^23^) and helix II (Ala^28^–Ala^38^) connected by a hinge region (Gly^24^–Ile^27^), with hydrophilic and hydrophobic terminals [42]. This peptide binds to the bacterial cell membranes and causes a loss of the electrochemical potential by halting ATP synthesis and active transmembrane transport [43,44]. Early studies [40,42] suggested that the C-terminal of the sarcotoxin IA penetrates the bacterial membrane, while the N-terminal interacts with the acidic phospholipids in the bacterial membrane, as the length of the helix I is thought to be identical to the thickness of this hydrophobic layer, resulting in perturbation of the membrane and inactivity of the bacteria. Thus, the binding of insect AMPs to microbial surfaces is a prerequisite for antimicrobial activity [45]. The majority of the insect AMPs are small and cationic [45], showing the effective activities against Gram-negative and occasionally Gram-positive bacteria [38,41]. Moreover, AMPs have minimal toxic and allergic side effects on humans and plants [38] and may potentially be applied in medicine and agriculture as safe antimicrobial agents [46]. However, there is no information in the current literature addressing the activity of the AMP sarcotoxin IA against *C*Las.

Based on these considerations, the need to have HLB-resistant/tolerant citrus cultivars to overcome the detrimental impact caused by this complex pathosystem is clear to the citrus industry. In this paper, we confirm the *Agrobacterium*-mediated transformation of Pera sweet orange mature plants with the *stx IA* gene and evaluate its reaction against *C*Las infection under protected conditions, insect-proof greenhouse, and the physiochemical quality of the fruit produced by these GMC lines.

## 2. Results

### 2.1. Integration and Expression of the stx IA Gene

The integration and stability of the *stx IA* gene that encodes the peptide with antimicrobial activity sarcotoxin IA in the genome of the experimental plants of the four Pera sweet orange transgenic lines were confirmed by conventional PCR (Figure 1). The 268 bp DNA fragment was amplified from all 96 plants with *C*Las-inoculated and non-inoculated transgenic lines STX-5, STX-11, STX-12, and STX-13 included in this study, confirming the *stx IA* gene integration in the genome of the plants (Figure 1). The non-transgenic control of Pera plants did not show any DNA amplification for this gene (Figure 1).

The expression of the *stx IA* gene was confirmed for all plants of the four lines: STX-5, STX-11, STX-12, and STX-13 (Figure 2). This expression was examined based on PCR amplification using cDNA synthesized from total RNA extracted from the Pera sweet orange transgenic plants. On the other hand, no amplification of the 268 bp DNA fragment was observed for the cDNA of the non-transgenic control plants (Figure 2). 

The quantitative detection of the *stx IA* gene product was examined based on the correlation between the number of copies of the sequence of the gene in the cDNA sample and the threshold cycle (Ct) values of the RT-qPCR test. The logarithmic standard curve of the PCR products was established based on eight serial dilutions, 1.38 × 10^9^ up to 1.76 × 10^4^, of the DNA extracted from the pST10 plasmid, containing the *stx IA* gene. The linear regression coefficients (R^2^) were 0.99, with an efficiency of 90.10, and a slope of –3.59 (Appendix A). These results indicate a high degree of accuracy over a wide range of the *stx IA* gene product concentration. The Ct values obtained to detect copies of the sequences that encode the *stx IA* gene ranged from Ct 10 with a maximum number of copies of 10^9^ up to Ct 27 with a minimum number of copies of 10^4^.

The plants of all four transgenic lines of Pera sweet orange constitutively expressed the *stx IA* gene (Figure 3), as previously revealed in the PCR test (Figure 2). However, the results obtained by RT-qPCR showed differences in the expression level of the *stx IA* gene among the transgenic lines (Appendix A). The plants of the STX-5 transgenic line showed the highest expression level of the *stx IA* gene, with 10.5 Log_10_ copies of the sequence encoded by the *stx IA* gene per gram of leaf tissue (Figure 3). On the other hand, the STX-13 transgenic plants had the lowest level of expression, with 9.2 Log_10_ copies of the sequence of the *stx IA* gene per gram of leaf tissue, while plants of the STX-11 and STX-12 lines showed intermediate levels of the gene expression. Furthermore, the non-transgenic control plants did not show any expression of this gene, as expected (Figure 3).

### 2.2. Huanglongbing (HLB) Evaluation of the CLas-Infected Plants

The plants of all Pera sweet orange transgenic lines expressing the *stx IA* gene exhibited normal growth over the experimental period, indicating that the expression of this gene did not have any deleterious effect on this sweet orange cultivar. However, the transgenic lines had distinct incidences of HLB, based on evaluations of the symptoms after *C*Las inoculations. Further, the *C*Las infection of the sweet orange plants was also examined by PCR tests.

The first HLB symptoms were observed in a few plants six months after inoculation. Mild symptoms were observed, including moderate blotchy mottle on the young and mature leaves. It is noteworthy that most plants, both transgenic and non-transgenic ones, did not show any typical HLB symptoms in the early stages after inoculation. However, an increase in HLB-symptomatic plants was noticed 60 months after the last *C*Las inoculation, for both transgenic and non-transgenic plants. Interestingly, the STX-11 and STX-5 transgenic plants exhibited reduced susceptibility to *C*Las over five years, scoring the lowest HLB cumulative incidence, around 50%. On the other hand, the non-transgenic control and the STX-13 plants had the highest cumulative incidence of HLB-affected plants, around 80%, as well as *C*Las-infected plants (Figure 4). These results were confirmed by PCR and RT-qPCR, as significant differences were observed among the transgenic lines for the HLB cumulative incidence. The STX-11 line exhibited the lowest incidence of *C*Las-infected plants, only 18%, while the non-transgenic control had the highest score, with 67% of diseased plants (Figure 4).

### 2.3. Evaluations of Fruit and Juice Quality

The effect of HLB on fruit quality was evident in our study, based on different parameters (Table 1). Fruit of the non-inoculated control plants was larger and heavier in comparison with those produced by the *C*Las-inoculated plants (HLB-affected fruit). Fruit from the *C*Las-inoculated control plants had a more than 15% reduction in weight and size compared with those produced by the non-inoculated plants. On the other hand, most of the fruit produced by the STX IA transgenic plants did not show any significant decrease in weight and size (Figure 5), except for the fruit of the STX-11 line which showed a slight decrease in weight due to *C*Las infection. In general, the fruit of the non-inoculated plants were rather oblong, typical of Pera sweet orange (Figure 5), as compared to the ones of *C*Las-inoculated plants that showed a misshapen appearance. The number of seeds was variable among the treatments. The fruit of STX-5, STX-11, and STX-12 lines had fewer fully developed seeds after *C*Las infection, while those of non-transgenic control plants had an increase in the number of normal seeds. It is worth mentioning that the fruit of the non-transgenic control plants had the highest number of aborted/underdeveloped seeds after *C*Las infection, typical of the fruit of HLB-affected plants. On the other hand, the fruit of all STX transgenic lines had a lower number of aborted seeds under *C*Las infection.

The fruit color attributes *L**, *a**, *b**, *C**, and CCI were significantly (*p* ≤ 0.05) affected by the CLas inoculation as well as by genetic transformation (Table 2). The coordinate *L** decreased in the peel of the fruit of the *C*Las-inoculated plants of the control and the STX-13 line, while the STX-11 fruit had an improvement in this color parameter. The highest *a** values, as higher values indicate a red color, were observed in the fruit of the non-inoculated plants of the lines STX-5 and STX-12. HLB-affected fruit showed an increase in this color parameter in the cases of the control and STX-11 line. Significant increases were also observed in the *b** values. A higher value indicates the less yellow and more green color for the HLB-affected fruit of the STX-11 and STX-13 lines, while no changes were found in the fruit of the control and the other transgenic lines. The coordinate chroma (**C*) increased in the fruit of the *C*Las-inoculated plants of the STX-11 and STX-12 lines, but did not change in any of the other transgenic lines and the control. Regarding the citrus color index (CCI), no significant variations were observed between the fruit of the non-inoculated and *C*Las-inoculated plants of the transgenic lines STX-11, STX-12, and STX-13. On the other hand, the HLB-affected fruit of the STX-5 line and control plants showed a reduction in this parameter. Further, lower positive CCI values indicate a less red and higher yellow color.

The juice color also showed variation. Juices of the fruit from the control, STX-5, and STX-13 *C*Las-inoculated plants had lower *L** values compared to the ones from non-inoculated plants, while it increased for the fruit of the STX-12 line. No such variation was observed for the STX-11 line. The *a** values of the juice samples were all negative. Fruit from non-inoculated plants produced juices with lower *a** than the juices of the HLB-affected fruit, except for the ones of the STX-12 line. In contrast, the HLB-affected fruit had juices of lower *b** than fruit produced by the non-inoculated plants, particularly in the case of the control and STX-5 line. These observations indicate that the juice of the fruit of non-inoculated plants was more yellow than that of the fruit from *C*Las-inoculated plants. A similar trend was observed for the **C* parameter, as the fruit from control plants had juice containing the lowest values for both non- and *C*Las-inoculated plants. The CCI calculated for the juice samples was lower in the non-inoculated plants rather than in the *C*Las-inoculated, except for the STX-12 line. The CCI values were all below 0, ranging from –5.85 to –3.62 for non-inoculated plants and between –5.08 and –4.21 among the *C*Las-inoculated ones, indicating a deep yellowish green color.

Significant interactions (*p* ≤ 0.001) were observed for juice content and chemical quality attributes of the Pera sweet orange fruit from non- and *C*Las-inoculated plants (Figure 6). The juice content of the fruit of the non-inoculated plants ranged from 35% up to 43%, which was significantly higher compared to the ones from the fruit of *C*Las-inoculated plants, which ranged from 41% down to 30%. Fruit harvested from control plants had the most contrasting juice contents, with 36% of juice in the fruit of the non-inoculated plants and only 30% in the ones of the *C*Las-inoculated plants. In contrast, the fruit of all the STX IA transgenic plants had more stable juice contents, with at least 35%. The acidity (TA) levels of the juices were significantly higher in the fruit of the non-inoculated plants, particularly for the STX-5, STX-11, and STX-13 transgenic lines. In general, the total acidity ranged from 0.6 up to 1.1 g of citric acid per 100 mL in the fruit from non-inoculated plants and between 0.5 and 0.7 g·100 mL^−1^ in the ones from *C*Las-inoculated plants. The juice of the non-inoculated STX-11 treatment had the highest content of citric acid, similar to the ones of the STX-5 line. No such differences were observed among the HLB-affected fruit of the transgenic lines for this parameter. The soluble solid contents (SSC) in the juices drastically dropped, from 12.6 down to 10.0 °Brix, when comparing the non-inoculated with the *C*Las-inoculated control treatments. Significant differences were also observed for the SSC levels in the fruit of the transgenic lines affected by HLB, except for the STX-12 line that did not show any response to the *C*Las inoculation. Regarding the SSC/TA ratio, the non-inoculated control plants had a higher maturity index than the *C*Las-inoculated, but no such trend was observed in fruit of the transgenic plants. In this way, the fruit of all STX IA transgenic plants had a higher SSC/TA ratio than the ones from the control for the HLB-affected fruit.

Significant interaction (*p* ≤ 0.001) was also observed regarding the sugar and organic acid composition of the juices extracted from the fruit of the transgenic and non-transgenic Pera sweet orange plants (Table 3 and Table 4). Sucrose was the most abundant sugar constituent in the juice samples, ranging from 114.7 g·L^–1^ for the STX-11 to 139.4 g·L^–1^ for the control of the non-inoculated (*C*Las^−^) plants. In contrast, the sucrose content decreased to 91.3–125.7 g·L^–1^ in juice from the *C*Las-inoculated (*C*Las^+^) plants. Similarly, the concentration of glucose and fructose in the juice was higher in the fruit of non-inoculated plants (glucose: 45.1–53.7 g·L^–1^; fructose: 45.1–55.5 g·L^–1^) than in the ones of *C*Las-inoculated plants (glucose: 32.0–52.8 g·L^–1^; fructose: 40.1–55.4 g·L^–1^), except for STX-11 and STX-12, which did not exhibit such tendency. Interestingly, these two transgenic lines had similar or even higher sugar content for the fruit of *C*Las-inoculated plants, ∑ = 222.4 and 234.0 g·L^–1^, respectively, indicating their improved performance in maintaining the sugar quality in the juice of HLB-affected fruit.

The organic acids citric, malic, succinic, and ascorbic were the main acids detected in the juice samples extracted from the fruit of non- and *C*Las-inoculated STX IA transgenic and non-transgenic Pera plants (Table 4). Among them, citric acid was the major organic acid, ranging from 22.7 up to 31.2 g·L^−1^ in the juice of the fruit from non-inoculated plants and from 19.0 to 24.2 g·L^−1^ in the fruit from *C*Las-inoculated plants. Moreover, citric acid represented more than 70% of the total organic acids in the samples. Interestingly, the *C*Las-inoculated plants produced fruit with a higher malic acid content, 2.87 to 6.34 g·L^−1^, than the non-inoculated ones, which showed a range of 2.22 up to 3.68 g·L^−1^, a two-fold increase in the case of the STX-11 and STX-13 lines (Table 4). Low variation in the succinic acid content was observed for the juice of the non-inoculated plants, ranging from 1.27 up to 1.99 g·L^−1^, though a high increase in the juice of the fruit of the *C*Las-inoculated plants, 0.47 up to 2.10 g·L^−1^. The amount of ascorbic acid decreased in the juice of most of the HLB-affected fruit, including the fruit from STX-5, STX-11, and ST-12 lines. In contrast, this organic acid increased significatively, from 0.98 and 1.49 g·L^−1^ to 1.30 and 1.61 g·L^−1^, in the control and STX-13 juice samples, respectively. In general, the total organic acids were relatively lower in the juice of HLB-affected plants than in the non-inoculated ones, except for the STX-12 line which showed an increase of 8.4% in the organic acids of the HLB-affected fruit.

### 2.4. Western Blot and Indirect ELISA Assays for the Sarcotoxin IA Detection

The sarcotoxin IA peptide was quantified in the juice extracts by an indirect ELISA assay. A significantly lower concentration of sarcotoxin IA peptide was found in the juice of the fruit of the STX IA transgenic line compared to the positive control containing the sarcotoxin IA synthetic peptide at 1.0 μg·mL^−1^ (Figure 7). The levels of sarcotoxin IA peptide accumulated in the juice extracted from the fruit of transgenic lines ranged from approximately 0.21 up to 0.31 μg·mL^−1^ of freshly squeezed juice. The juice of the STX-5 and STX-12 lines had the lowest accumulation of the sarcotoxin IA peptide. On the other hand, only the juice of these two STX IA transgenic lines showed the presence of the sarcotoxin IA protein in the juice, as revealed by the Western blot analysis (Figure 8). The juice of all other transgenic lines did not show any presence of this protein. 

## 3. Discussion

### 3.1. Reaction of the STX IA Transgenic Plants to CLas Infection

Over the last two decades, HLB has severely affected citrus production globally, particularly in the citrus-producing areas where the disease has become established, including some regions of Brazil, the United States, Mexico, and Argentina. Losses in the HLB-affected orchards are high, as the infected plants show limited vegetative growth, massive fruit drop, reduced yield, and economic plant death a few years after bacterial infection [8,13,30]. Several studies have been carried out all over the world to overcome this problem, searching for measures to improve citrus management under the presence of HLB [20].

A shortage of *Citrus* spp. with a certain level of HLB resistance, associated with the difficulty of using conventional breeding approaches, makes genetic transformation the most appropriate technique to incorporate traits of interest in high-demand citrus cultivars [37,47]. Genetic transformation enables a precise and controlled insertion of a desirable agronomic trait in a relatively short period, particularly in wood perennial species [39]. “Genetically modified citrus” (GMC)-expressing genes that encode antimicrobial peptides (AMPs) have shown promising results to increase resistance against bacterial diseases [33,48,49]. Additionally, transgenic Pera sweet orange plants encoding the *stx IA* gene have increased resistance against bacterial diseases, such as citrus canker, under controlled conditions [37,39].

Therefore, this study was conducted to investigate the potential of transgenic Pera sweet orange transformed with the *stx IA* gene [37] against *C*Las, the most widespread HLB bacterium that infects citrus plants. The first step involved the confirmation of the integration and the stability of the *stx IA* gene in all Pera sweet orange plants included in this study, as well as the *stx IA* gene expression. All plants of the four transgenic lines of Pera showed the integration and stability of the *stx IA* gene in the plant genome (Figure 1). These findings confirmed the transformation of these transgene lines with the *stx IA* gene, as previously reported [37]. However, the results obtained here not only corroborate the previous report on the insertion of this specific gene into the citrus genome, but also confirm the integration and stability of the *stx IA* gene in the Pera sweet orange plants. Furthermore, the constitutive expression of the *stx IA* gene and the consequent accumulation of the sarcotoxin IA peptide in the plant tissues did not cause any deleterious effect on their development.

The quantification of the *stx IA* gene expression by RT–qPCR through cDNA synthesized from total mRNA revealed significant differences in the expression of this gene by the four independent transgenic lines (Figure 3; Appendix A). Plants of the STX-5 line showed the highest level of *stx IA* gene expression, differing significantly from the expression observed in STX-13 plants (Figure 3). On the other hand, plants of the other two transgenic lines showed intermediate levels of the *stx IA* gene expression (Figure 3). Differences in the expression of genes encoding antimicrobial peptides had been previously reported [50], but not for citrus plants expressing the *stx IA* gene. However, variations in the levels of the *stx IA* gene expression did not indicate a relationship with the number of transgene copy insertions, as all STX lines included in this study have just one copy of the *stx IA* gene in their genome [37]. Similar results were also found for other transgenic citrus plants involving different gene constructions [35,50]. Variations in transgenes expression in plants containing the same gene construction may be related to the position where the gene was inserted in the genome [50,51].

The plant reaction against *C*Las infection was investigated in an insect-proof greenhouse. The first HLB symptoms developed six months after *C*Las inoculation by grafting the diseased buds. The symptoms consisted of light mottle in young and mature leaves. However, not all transgenic and non-transgenic control plants showed HLB symptoms during the entire experimental period. However, the control and the STX-13 plants showed the highest HLB incidence (Figure 4), while the STX-5 plants, which had the highest *stx IA* gene expression (Figure 3), exhibited the lowest HLB incidence, similarly to the STX-11 line (Figure 4). Unlike other citrus diseases, such as citrus canker, which only causes localized infections/lesions and has been the target of many studies with transgenic plants expressing genes that encode AMPs [37,49,50], HLB is a systemic insect-vectored bacterial disease showing complex systemic movement within the plant [52]. Further, the *C*Las bacterium is restricted to the phloem vessels of the citrus plants moving and redistributing all over the citrus plant [53]. The *C*Las distribution in the citrus plant is not uniform [52] and relies on several factors, involving environmental conditions such as temperature and light, plant age, citrus species, the nutritional condition of the plant, and concentration of the bacterium at the infection time [54,55,56]. These factors may regulate the interval between infection and the expression of the symptoms, as well as the severity of these symptoms [4,15,57,58,59]. Thus, these factors may have influenced the manifestation of the HLB symptoms in our study.

In addition to the HLB symptoms, the plant reaction to *C*Las was evaluated based on the quantification of the bacterium by PCR techniques. Based on conventional PCR tests, all transgenic and non-transgenic lines were infected by the *C*Las bacterium, though they had variations in the level of infection (Figure 4). The highest number of plants testing positive for the presence of *C*Las was observed in the non-transgenic control plants (Figure 4). However, these results by conventional PCR may be biased due to *C*Las uneven distribution in the different organs and tissues of the plant [4,18,53], as previously discussed.

The high number of plants infected with *C*Las revealed by the qPCR test is certainly related to the sensitivity of this technique [60]. Although the transgenic Pera sweet orange plants were not immune to *C*Las infection, some lines were less susceptible to the infection by the HLB pathogen. STX-11 plants expressed the lowest *C*Las titer, differing substantially from the non-transgenic control. Similar results were observed by Kobayashi et al. [37] with Pera plants of the same transgenic lines for infection by *Xanthomonas citri* subsp. *citri* (*Xcc*), the causal agent of the citrus canker disease. The *stx IA* gene expression in other plant species also increased the resistance to some diseases caused by plant pathogenic bacteria, including *Erwinia carotovora* subsp. *carotovora* and *Pseudomonas syringae* pv. *tabaci* [61]. Citrus plants expressing genes encoding other AMPs, such as cecropin B, have also shown improved tolerance to HLB, but none of the tested transgenic plants were immune to *C*Las [62].

Based on the quantification results of *C*Las by qPCR, bacterial titers varied considerably in the citrus plants of the different transgenic lines (Appendix A). The comparison of the bacterial titers in the different citrus genotypes revealed that, in many cases, the cumulated levels of *C*Las do not necessarily show a high correlation with the severity of the symptoms expressed by the host plant [18]. It should be pointed out that *C*Las does not necessarily have the same distribution and concentration in the different host plants. High bacterial titers were found in symptomatic *C. macrophylla* tissue, with 10.4 copies per gram of tissue [18]. However, asymptomatic tissues occasionally tested positive for the presence of the bacterium, with 8.26 copies of *C*Las per gram of tissue [18]. This pattern of infection makes it difficult to predict where *C*Las is located when HLB symptoms are absent.

*C*Lam titer may range from 5.6 up to 8.0 copies per gram of tissue in sweet orange leaves with mottle symptoms of HLB [63]. In our study, the results of qPCR showed variation between 6.6 and 8.7 copies of *C*Las per gram of tissue (Appendix A). It is noteworthy that *C*Las can reach a higher population in symptomatic leaves than *C*Lam [63]. However, STX-5 plants, which showed high expression of the *stx IA* gene, also exhibited the highest number of copies of the *C*Las bacterium in the qPCR test (Appendix A). The population of endophytic microorganism antagonists on *C*Las could have been altered by the expression of the AMP sarcotoxin IA. Furthermore, previous studies revealed that infections by HLB bacteria not only affect the citrus plant but also have significant effects on the structure and composition of the endophytic bacterial community associated with the citrus plant [64].

### 3.2. Quality of the Fruit Produced by the STX IA Transgenic Plants under CLas Infection

Fruit quality is modulated by several factors, including soil–climate conditions, hormonal crosstalk, scion–rootstock interaction, nutritional imbalance, disease incidence, insect pest attacks, water relation, and salinity, among others. In our study, we observed that HLB affected the fruit quality of the STX IA transgenic lines of Pera sweet orange differently. In general, the fruit from *C*Las-inoculated plants was smaller and lighter, characteristic of HLB-affected fruit [26]. However, this detrimental effect was minimized or even did not show up in fruit from the STX IA transgenic lines (Table 1, Table 2, Table 3 and Table 4; Figure 6).

HLB-symptomatic fruit is usually depreciated and sometimes unmarketable because it becomes smaller, greener with color inversion, and asymmetric (misshapen fruit), and also contains small brownish aborted seeds [11,65,66]. Nehela and Killiny [7] suggested that the typical HLB misshapen caused by the *C*Las infection in citrus fruit is related to complex molecular, metabolic, and anatomical changes governed by phytohormones. The analysis of the spatial distribution of the phytohormones indole-3-acetic acid (IAA) and abscisic acid (ABA) in the HLB-symptomatic fruit of Valencia sweet orange has generated new insights into the development of misshapen areas in citrus fruit [17]. These authors reported a two-fold increase in IAA content in the flavedo tissue isolated from misshapen sections of HLB-symptomatic fruit, compared to the normal-shaped sections of the fruit. However, the number of cells was reduced in the misshapen areas [17]. Due to the role of IAA in cell enlargement, fewer and larger parenchyma cells in one targeted side of the symptomatic fruit may explain the asymmetric growth pattern, resulting in a misshapen fruit [7,17]. Rosales and Burns [17] also observed that the ABA concentration was not equally distributed in the flavedo, by comparing the peduncle and style ends of the fruit. Furthermore, the ethylene levels were lower in the HLB-symptomatic fruit than in the non-infected ones, which may have played a key role in some physiological processes during fruit development and ripening. All these reports on citrus fruit size and shape altered by the *C*Las infection were also observed in our study. Non-transgenic control plants produced more misshapen fruit of smaller size than the plants of the STX IA transgenic lines, under controlled conditions. The changes in the *C*Las-infected fruit led to abnormal growth development that regulated the fruit shape to a less oblate, typical of sweet oranges. The size and weight of the Pera sweet orange fruit found in our study were similar to those reported for this cultivar under field conditions (Figure 5) [67]. Further investigation is certainly needed to evaluate the performance of the STX IA transgenic plants under field conditions.

Sweet orange fruit is colored by a natural process during ripening. The degradation of chlorophyll in the flavedo cells leads to a series of changes in fruit coloration [68]. As the fruit matures, the greenish color (chloroplast organelles) gives room to the yellowish color (chromoplast organelles) due to the biosynthesis of pigments such as carotenoids and flavonoids, which rules the color development in sweet oranges [69,70,71]. External factors, such as temperature and natural light, also have a broad contribution to color change in the citrus fruit [70]. Based on our findings, fruit from non-inoculated plants had a brighter color than the HLB-affected fruit, indicated by the coordinate *L**. These results were confirmed by the citrus color index (CCI), as a lower value was recorded in the control of the HLB-affected fruit. Interestingly, the majority of the STX IA transgenic fruit did not differ significantly (*p ≤* 0.05) regarding CCI despite *C*Las infections, except for the STX-5 fruit that showed lower CCI for the HLB-affected compared to the non-inoculated ones (Table 2).

The *a** values of the samples were all above 3.5, suggesting a green-less color. Although the STX-5 samples exhibited a reduction in the *a** values in the HLB-affected fruit, the samples of all other STX transgenic lines remained stable or even had a red improvement. On the other hand, the *b** values were significantly increased in the fruit of the STX-11 and STX-12 lines, indicating an advancement in the yellow color. This trend was also observed for the coordinate *C**, with a brighter color for the fruit of these transgenic lines.

Regarding the juice color, which is an important attribute for the juice processing industry, a slight variation among the treatments was also found for the three-dimensional CIE Lab coordinates. Juices of the non-inoculated plants were usually brighter than the ones of the HLB-affected fruit, with a muted color. This was indicated by the coordinate *L**. However, they were more yellow-colored than the juice of the HLB-affected fruit, expressed by the *a** and *b** values and confirmed by the CCI. Despite these variations, the results indicated that the juice color profile of the Pera sweet orange fruit of all the STX IA transgenic lines was not affected by the genetic transformation.

The juice content in the fruit ranged from 35% up to 43% for the STX IA transgenic lines for the samples of the non- and *C*Las-inoculated plants. This variation is within the acceptable level of ≥33%, established by the citrus industry [69,71,72,73,74]. |The juice content in sweet oranges plays a key role in fruit quality, as it is involved in the contents of soluble solids and acid levels by mere dilution. Juice content relies on water relations [75] and the ripening process, as it increases when the fruit matures, until full maturity, and decreases afterward [71,74].

The chemical quality parameters of the juice are the main tool to determine the maturation stage of sweet orange, as fruit size and color are not reliable attributes for this purpose. The sugar contents in sweet oranges progressively increase during fruit growth and maturation, though they are dependent on the photosynthetic machinery that also relies on the temperature and light intensity [69]. The soluble solid content (SSC), expressed as degree Brix, is a measure of the sweetness level of the fruit juice, which is related to the sugar content in the fruit at a particular maturation stage [71]. The SSC measured in the juices extracted from HLB-affected fruit was lower than the ones from non-inoculated plants, corroborating previous reports [76,77]. It is worth mentioning that the juice of the non-transgenic fruit had substantial variation in SSC when compared with the fruit of non-and *C*Las-inoculated plants. In contrast, a minor variation in SSC was observed for the fruit of non- and *C*Las-inoculated transgenic lines. The sugar profile of the juice samples has sucrose (Table 3), the most abundant sugar constituent of the sweet orange juices [66,78,79]. Sugar content decreases in the juices of HLB-affected fruit for some transgenic lines, similar to the SSC. However, an increase in sugar content in the HLB-affected fruit was observed for the STX-11 and STX-12 lines (Table 3). Similar results were found in the composition of glucose and fructose, but at a lower frequency compared to sucrose.

The total acidity in sweet orange juices is mainly related to citric acid, the main organic acid present in these fruits (Table 4) [80,81]. Organic acids are formed in the oxidative tricarboxylic acid (TCA) cycle, known as Krebs or the citric acid cycle, by the mitochondrial pathway in the juice cells [82]. The organic acid level in sweet oranges usually decreases as the sugar constituent increases [71]. The acidity level measured in this study ranged from the lowest 0.5 up to 1.1 g of citric acid per 100 mL of juice. The fruit juice of non-inoculated plants was usually more acid than the juice extracted from the fruit of *C*Las-inoculated plants, particularly those of the STX-5 and STX-11 transgenic lines, which had the highest levels, around 1.0% (Figure 6; Table 4). These findings contrast with the results reported for Hamlin and Valencia sweet orange juices of *C*Las-positive plants, which had higher acidity levels in the HLB-affected fruit than in the non-affected ones [66,77,79,83]. These differences in acidity levels observed between fruit juices of non- and *C*Las-inoculated plants were mainly related to the reduction in the citric acid concentration after *C*Las infection, for most of the STX transgenic lines and the non-transgenic control plants (Table 4). It confirms that the citric acid rules the total acidity in the sweet orange juice even under bacterial infestation. This pattern was also reported in a previous study on Valencia sweet orange juice [76,84]. Interestingly, our findings demonstrated that the malic acid concentration promptly increased in the HLB-affected fruit, while the succinic and ascorbic acid levels remained stable (Table 4). Despite these variations, the total acidity recorded in our study was appropriate for sweet orange commercialization, as no disruption was observed in the acid grading which reached acceptable levels of palatability and consumption [69,71,74].

The sugar–acid (SSC/TA) ratio is the most acceptable indicator that regulates the sweet orange fruit maturation and harvest season [69,71,74]. The sweetness of the sugars and the sourness of the organic acids compete for the same receptor sites in the human tongue, which drives the sugar–acid ratio to such a relevant parameter in determining the juice quality of sweet oranges [85]. Indeed, the sweet orange fruit is considered marketable when this ratio reaches the minimum standard. This parameter is imposed by the fresh and industrial markets [72,73], which is the basis of sensory perception and consumer preference. According to the international standards ruled by the fresh citrus market, sweet orange juices must have a ratio of at least 6.5:1 (SSC/TA) [72], while the Brazilian citrus fresh market demands 9.5:1 as the minimal sugar–acid ratio [73]. In general, sweet orange juice scoring SSC/TA ratios between 7:1 and 9:1 are considered acceptable for commercialization in several countries [71,74]. On the other hand, the citrus industry requires a sugar–acid ratio at a higher grade, between 12 and 18, to start processing fruit for the frozen concentrated orange juice (FCOJ) and not-from-concentrated (NFC) orange juices [86]. However, changes in industrial processing have been occurring to address the lack and disorders caused by the HLB in the citrus orchards, particularly those related to fruit quality, which have affected the orange juice production around the world. Currently, the juice processing industries are blending low-quality (HLB-affected) with high-quality (non-HLB-affected) sweet orange juices to meet industrial standards. However, no such procedure can be applied to the fresh fruit market [26]. Under this scenario, the sugar–acid ratio observed in our study inferred that the fruit of all STX IA transgenic lines, non- and *C*Las-inoculated plants, produced fruit with a superior ratio than the minimum established by the fresh fruit market. Now, when we look at the industrial requirements, all HLB-affected transgenic fruit had sugar–acid ratios higher than 12, attending the industrial standards. However, fruit from non-inoculated STX-5 and STX-11 lines did not reach this threshold, because their juices were significantly higher in total acidity, though the SSC met the marketable standard (Figure 6).

The sarcotoxin IA concentration in the juice extracts was determined by the protein-blotting test. Extracts of the juices from STX-11 and STX-12 fruit, belonging to the transgenic lines that scored the lowest *stx IA* gene expression (Figure 3), did not reveal any sarcotoxin IA protein residue by immunoblotting. Thus, these transgenic lines produce low amounts of the sarcotoxin IA peptide that could scarcely be detected by protein blot analysis, as mentioned by Okamoto et al. [87], probably due to the instability of the short peptides in plant cells. On the other hand, protein traces were detected on the STX-5 and STX-13 lines, the ones that scored the highest *stx IA* gene expression (Figure 3).

However, the presence of the sarcotoxin IA peptide was evident in the juices of all STX IA transgenic lines, even in low concentrations (0.21–0.31 μg·mL^−1^ of freshly squeezed juice), as determined by the indirect ELISA test. The accumulation of this peptide did not cause any deleterious effects on fruit or juice composition, similar to previous report on plants of the same sweet orange cultivar [37].

Moreover, previous *in vitro* study has indicated that this AMP has a selective antibacterial activity, suppressing the growth of harmful human bacteria including *Clostridium ramosum*, *C. paraputrificum*, and *Escherichia coli* O157, and having no detrimental effect on healthy intestinal microbiota [88]. This study also suggested that sarcotoxin IA-containing crops might also exhibit enhanced resistance to contamination by food poisoning bacteria during harvesting and transportation, reducing the need for agricultural chemicals and being a safe source of food for humans. However, further investigations are still pending to understand the antimicrobial activity of the AMP sarcotoxin IA against plant bacterial agents such as *C*Las.

Taken together, genetic transformation to overcome disease threats is a laborious and time-consuming solution for long-term perennial crops such as citrus, and must include different sources of tolerance/resistance inducers, in order to prevent a mutation in the pathogen [89]. However, concerns are still widespread among the community regarding the commercial cultivation of “genetically modified crops”, even under scientific evidence that the improved crop is safe while increasing resistance against biotic and abiotic stressors, yield, and nutritional quality, besides reducing environmental harm [90]. On the other hand, it is necessary to fully understand the effect of the overexpressed insect-derived AMPs in transgenic plants on human health and food security before commercial cultivation [91], as in the case of the sarcotoxin IA. Thus, the importance of new investigations on genetic improvement is clear, as the development of disease-resistant citrus may have a positive effect on citrus protection and production [90].

## 4. Materials and Methods

### 4.1. Plant Materials and Transformation Construct

Four sarcotoxin IA (STX IA) transgenic lines of Pera sweet orange clone Rio (*Citrus*
*× sinensis* (L.) Osbeck), the most cultivated sweet orange in the Brazilian orchards [92], were included in this study, i.e., STX-5, STX-11, STX-12, and STX-13. The transgenic lines were previously transformed and present the *stx IA* gene isolated from the flesh fly (*Sarcophaga peregrina*) that encodes the peptide sarcotoxin IA, which has antimicrobial activity [37]. Plant transformations were performed through a disarmed *Agrobacterium tumefaciens* strain EHA-105 carrying the pST10 plasmid [87] at the Laboratory of Biotechnology of the Instituto de Desenvolvimento Rural do Paraná—IAPAR/Emater (IDR-Paraná) in Londrina, state of Paraná, Brazil [37]. This plasmid contains the *stx IA* gene joined to a signal peptide from *Nicotiana tabacum* PR1a, under the control of the double-enhanced strong constitutive 35S promoter obtained from *Cauliflower mosaic virus* (CaMV) and the omega locus from *Tobacco mosaic virus* (TMV) [37,39]. The pST10 plasmid also contains the selective marker *nptII* gene under the control of the *nopaline synthase* (*nos*) promoter and terminator sequences [37]. All non-transgenic control and transgenic STX lines of Pera sweet orange nursery plants were grafted on Rangpur lime (*C.*
*× limonia* (L.) Osbeck).

### 4.2. Plant Cultivation

Twenty-four plants of each treatment, i.e., STX-5, STX-11, STX-12, and STX-13 transgenic lines and non-transgenic, were cultivated in eight-liter pots containing sterilized commercial substrate composed of ground *Pinus* spp. bark (Plantmax^®^, Eucatex Química e Mineral Ltd., São Paulo, SP, Brazil). The potted plants were kept in a semi-climate-controlled insect-proof greenhouse with air filtration on the outside. Air temperatures ranged from 20 to 35 °C throughout the day, all over the experimental period. Plant management included regular irrigation, plant-soil nutrition, insect pest and disease control, and pruning, among other practices.

### 4.3. Confirmation of the Integration and Expression of the stx IA Gene

For confirmation of the integration, stability, and expression of the *stx IA* gene in all experimental plants of the transgenic lines, the total genomic DNA was extracted from 500 mg leaf midrib using the Murray and Thompson [93] protocol. The complementary DNA (cDNA) was synthesized from the total RNA extracted from 100 mg of leaf tissue samples using the High-Capacity cDNA Reverse Transcription Kit (Applied Biosystems^®^, Foster City, CA, USA). The total genomic DNA and the cDNA were used to perform the molecular analyses with specific primers in the polymerase chain reaction (PCR). The primers set STX IA F and STX IA R were used for amplification of a 268 bp fragment of the *stx IA* gene [37].

PCR reaction was performed in 20 µL of the reaction mixture containing 2.0 µL of PCR buffer (1×), 0.8 µL of dNTP (5 mM), 0.6 µL of MgCl_2_ (50 mM), 1.0 µL of each specific primer, 0.2 µL of recombinant *Taq* DNA polymerase (5 UµL^−1^), 13.4 µL of ultrapure water, and 1 µL of DNA or cDNA for the integration and expression of the *stx IA* gene, respectively. The negative control was included in each PCR, as the DNA or the cDNA was replaced by ultrapure water. The positive control for the *stx IA* gene was also included in each PCR.

A thermal cycler (Veriti^TM^ 96-Well, Applied Biosystems^®^, Waltham, MA, USA) with the following program was used for DNA amplification: 30 cycles at 94 °C for 60 s, 60 °C for 45 s, 72 °C for 45 s, and a final extension at 72 °C for 5 min. PCR products were electrophoresed in 1.0% (*w*/*v*) agarose gel and visualized with SYBR™ gold nucleic acid gel Stain (ThermoFisher Scientific, Carlsbad, CA, USA). Images were visualized by a photo documenter (L-PIX EX, Loccus do Brasil Ltd., Cotia, SP, Brazil) under UV light.

The cDNA was also used to quantify the expression of the *stx IA* gene by the reverse transcription quantitative PCR (RT-qPCR) assay in the real-time PCR system (ViiA^TM^ 7, Applied Biosystems^®^, Waltham, MA, USA). The RT-qPCR reactions were prepared with the primers set STX IA F/R (Appendix A) [37]. The real-time PCR system was set up to the standard amplification of 2 min at 50 °C and 2 min at 95 °C, followed by 40 cycles of 15 s at 60 °C and 30 s at 60 °C. At the end of the reactions, the melting point of the amplicons was determined by subjecting the samples to 95 °C for 15 s and gradually heating from 60 to 95 °C at a rate of 0.3 °C·s^–1^ and holding at 60 °C for 15 s. To build the logarithmic standard curve for absolute quantification of the *stx IA* gene expression, eight serial dilutions of a plasmid pST10 (3038 bp) DNA-derived amplicon [93] were developed (5, 1, 0.2, 0.04, 0.008, 0.0016, 0.00032, and 0.000064 ng·µL^−1^). Each dilution point was performed in triplicate. Ct values were obtained for each dilution and used to quantify the number of equivalent copies of the *stx IA* gene sequence, determined by the equation generated from the log curve (Appendix A). Results were expressed in Log_10_ at 1.0 μL of RNA. RT-qPCR reactions were submitted to similar conditions and normalized by the ROX passive reference dye signal, in order to correct reading fluctuations due to variations in volume and evaporation throughout the reaction.

### 4.4. ‘Candidatus Liberibacter asiaticus’ (CLas) Inoculation

HLB-symptomatic branches from *Citrus* spp. plants were collected at the Experimental Station of Londrina (IDR-Paraná) for *C*Las inoculation. The presence of the bacterium *C*Las in the plant material was confirmed by PCR. Total genomic DNA was extracted from 500 mg of the leaf midribs [92] and subjected to PCR amplification as described below.

*C*Las-infected buds, 1.5 cm in length, were grafted on the STX IA transgenic lines and non-transgenic control Pera nursery plants, using the inverted T budding method. Two *C*Las-infected buds were grafted on each plant. A second *C*Las inoculation was performed six months later, using the same procedure described above. Fifteen days after the last inoculation, all plants were pruned to stimulate new flushing growth and consequent bacterial redistribution. Twelve plants per treatment, i.e., STX IA transgenic lines and non-transgenic, were graft-inoculated with *C*Las, while the other twelve plants per treatment were maintained non-inoculated and used for fruit production under protected conditions.

### 4.5. Huanglongbing (HLB) Evaluation of the CLas-Infected Plants

The transgenic and non-transgenic Pera plants were evaluated at one and six years after *C*Las inoculation for the presence of HLB symptoms. The results were expressed as the cumulative incidence of HLB, which was related to the number of symptomatic plants over the evaluation period (2016–2021). Visual inspections were performed based on the presence of typical HLB symptoms in the leaves and fruit of citrus plants [4].

Molecular analyses by conventional PCR were also performed to confirm the *C*Las infection in the inoculated plants. Total genomic DNA was extracted as previously described [93] and subjected to PCR amplification using the primer sets A2/J5 and Oi1/Oi2c (Appendix A) that amplify a fragment of 703 bp of the β-operon locus of ribosomal proteins and 1160 bp of the 16s rDNA locus, respectively [9,94]. PCR reactions were performed in a thermal cycler (Veriti^TM^ 96-Well, Applied Biosystems, Waltham, MA, USA). For the A2/J5 primers set, the thermal cycler was set up to 35 cycles at 92 °C for 20 s, 62 °C for 20 s, and 72 °C for 45 s [94], while there were 35 cycles at 92 °C for 40 s and 72 °C for 90 s for the Oi1/Oi2c primers [9].

Following amplification, aliquots of each PCR reaction mixture were analyzed by electrophoresis in 1.0% (*w*/*v*) agarose gel. PCR analyses were performed during the evaluated period, 2016 through 2021, to examine the evolution of the *C*Las infection in the STX IA transgenic and non-transgenic plants.

The quantitative determination of *C*Las in the Pera sweet orange transgenic lines was performed by qPCR. The DNA for the qPCR was the same extracted for the conventional PCR, as described previously. The concentration and purity of the DNA were determined by spectrophotometry using a NanoDrop^®^ 1000 (NanoDrop Technologies, Thermo Fisher Scientific Inc., Wilmington, DE, USA). The final DNA concentration was adjusted to 100 ng·µL^−1^.

qPCR was performed in 15 µL of reaction mixture containing 7.5 µL of Master Mix (2×) (*Taq*Man^®^ Universal PCR Master Mix kit, Applied Biosystems™), 0.37 µL of each target primer at 100 nM (HLBas and HLBr), 0.45 µL of each internal control probes (HLBaspr and COXp), 0.37 of each internal control primers (COXf and COXr), 3.0 µL of DNA at 100 ng·µL^−1^, and 2.1 µL of ultrapure water. Primers and *Taq*Man^®^ probes that amplify the 16s region of the *C*Las rDNA and cytochrome oxidase (COX) protein were designed according to Li et al. [95].

The RT-qPCR system was set up for amplifications at 60 °C for 2 min and 95 °C for 10 min, followed by 40 cycles at 95 °C for 15 s and 58 °C for 60 s. qPCR reactions were performed in triplicate, including a positive control containing *C*Las DNA from an infected plant, a negative control containing DNA from an uninfected plant, a blank reaction containing ultrapure water, and a plasmid pGEM^®^-T easy vector (Promega, Madison, WI, USA) (3015 bp) containing the *C*Las *16s rDNA* gene (1160 bp), obtained from Oi1 and Oi2c primer amplification [9]. The mitochondrial *cytochrome oxidase* (*COX*) gene from *Citrus* spp. plants was used as an internal positive control in the reactions to determine the quality of the extracted DNA and the qPCR reactions.

The quantification of the *C*Las cells was based on the standard curve developed by seven serial dilutions at 100 ng·µL^−1^ of a plasmid DNA. Ct values were used to calculate the regression equation to determine the equivalent number of *C*Las in the citrus extracts: y = –1.505x + 45.415. The calculated results were expressed in Log_10_ of *C*Las cells per milligram of tissue. Only samples with Ct ≤ 34 were considered *C*Las-positive.

### 4.6. Evaluations of the Fruit and Juice Quality

#### 4.6.1. Fruit Harvest

Fruit quality evaluations were assessed by comparing fruit harvested from the non- and CLas-inoculated potted plants. Twenty-five fruits were randomly collected from twelve plants per treatment and divided into five replicates of five fruits each.

#### 4.6.2. Physicochemical Analysis

Fruit length and diameter were measured with a digital Vernier caliper (ABS, Mitutoyo, Kawasaki, Kanagawa, Japan) and weighed on a digital scale. The fruit shape index was calculated based on the relationship between fruit length and diameter.

Peel and juice colors were measured using a portable chroma meter (CR-400, Konica Minolta, Tokyo, Osaka, Japan) and the CIE Lab color system [96]. The device was calibrated before color assessments with a white tile. The fruit color was determined by readings at four equidistant points from the equatorial circumference of every single fruit, whereas the juice color measurements were taken on a 10 mL cuvette filled with freshly squeezed juice. The lightness (*L**) value ranges from 0 (black) to 100 (white), in which higher values indicate lighter color intensity; the positive coordinate *a** represents the red color and the negative coordinate indicates the green color, while positive and negative coordinates *b** represent the yellow and blue colors, respectively [96]. The chroma (*C**) and the citrus color index (CCI) were also calculated according to the following equations:(1)$$C^{*}=\sqrt{a^{*2}+b^{*2}}$$
where *C** = chroma, *a** = red–green color value, and *b** = yellow–blue color value [96].
(2)$$\mathrm{CCI}=\frac{1000\times a^{*}}{L^{*}\times b^{*}}$$
where CCI = citrus color index, *a** = red–green color value, *b** = yellow–blue color value, *L** = lightness [97].

The *C** values express the intensity or the saturation of the color, while the CCI is a comprehensive indicator for color impression with positive values for red; negative values for blue–green; and 0 for an intermediate mixture of red, yellow, blue, and green [98].

Fruit samples were juiced using an extractor (Croydon^®^, Duque de Caxias, RJ, Brazil). The juice content was determined based on the relationship between juice weight and fruit weight and expressed in percentage (%). Soluble solids content (SSC) was assessed with a digital refractometer (PAL-3, Atago^®^, Tokyo, Kantō, Japan) in 0.3 mL of undiluted juice. The values were corrected to 20 °C and the results were expressed in Brix units. Titratable acidity (TA) was determined in 25 mL of diluted juice (juice:distilled water; 1:3) and 0.1 N NaOH in a titrator (TitroLine^®^ easy, Schott Instruments GmbH, Mainz, Rhineland-Palatinate, Germany), and expressed in grams of citric acid per 100 mL of juice (g·100 mL^−1^) [99]. The SSC/TA ratio was calculated to determine the fruit maturity.

#### 4.6.3. Determination of Sugars and Organic Acids Contents

Next, 5 mL of freshly squeezed juice was diluted in 10 mL of ultra-pure water (Milli-Q), vortexed for 1 min, and centrifuged at 1250× *g* (ultracentrifuge CP100WX, Hitachi Ltd., Tokyo, Japan) for 10 min. The supernatant was filtered (0.22 μm polyester membrane, Sartorius Stedim Biotech, Göttingen, Germany), transferred to 2.0 mL vials (Agilent Technologies, Walldorf, Germany), and used to determine and quantify the organic acids and carbohydrates (sugars). Chromatographic analyses were performed through an HPLC instrumental system (LC-20 A, Shimadzu Corp., Kyoto, Japan), equipped with a high-pressure pump system (LC-20AT), automatic injector (SIL-20AC HT), refractive index detector (RID-10A), photodiode array (SPD-M20A), column oven (CTO-20A), and control module (CBM-20A). Data collection and integration of the chromatographic peaks were processed using the LC Solutions software (Shimadzu Corp., Kyoto, Japan).

Samples were injected into the HPLC system. The injection volume was 20 μL. Organic acids were analyzed in a Luna C_18_ column (250 × 4.6 mm, particle size 5 μm, Phenomenex, Torrance, CA, USA). The mobile phase consisted of a 25 mM sodium phosphate buffer solution, and the pH was adjusted to 2.5 with phosphoric acid, at a flow of 1.0 mL·min^−1^. The column was maintained at 40 °C. Detections were performed through refractive index and photodiode array, programmed at 215 nm using a scan mode between 200 and 400 nm. Carbohydrates were analyzed through an Aminex HPX-87P column (300 × 7.8 mm, Bio-Rad Laboratories, Inc., Hercules, CA, USA) maintained at 80 °C. The mixture of acetonitrile and water (75:25, *v*/*v*) was used as a mobile phase at a flow rate of 1.0 mL·min^−1^, according to Pauli et al. [100].

### 4.7. Western Blot and Indirect ELISA Assays for the Sarcotoxin IA Detection

The crude protein extracts for Western blot analysis were obtained from freshly squeezed orange juice. Briefly, 40 μg of the total protein per sample was fractionated to 13.5% (*w*/*v*) SDS-PAGE and electroblotted on polyvinylidene difluoride (PVDF) nylon membrane (Millipore Corp., Burlington, MA, USA), using a semi-dry transfer system (Bio-Rad Laboratories, Hercules, CA, USA). Immunodetection was performed using 1.0 μg·mL^−1^ polyclonal rabbit anti-sarcotoxin IA antibody as the primary antibody and the alkaline phosphatase-conjugated anti-rabbit IgG as a secondary antibody (Sigma-Aldrich Ltd., St. Louis, MO, USA), diluted to 1:10,000. Synthesized sarcotoxin IA peptide was used as the standard for the relative quantification of sarcotoxin IA protein in the juice samples. Images were taken from the PVDF membrane and recorded on X-ray film.

The detection of the sarcotoxin IA peptide in the juice samples was also confirmed by the ELISA assay. Juice samples were diluted 1:1 (*v*/*v*) in a 50 mm sodium carbonate buffer (pH 9.6) during the microplate sensitization. After blocking, the polyclonal anti-sarcotoxin IA antibody was incubated at 1.0 μg·mL^−1^ and the alkaline phosphatase-conjugated anti-rabbit IgG was added as a secondary antibody (Sigma-Aldrich Ltd., St. Louis, MO, USA) at 1:10,000. After washing the plate, 100 μL of TMB was added to each well for color development at 37 °C for 10 min. The reactions were stopped by adding 50 μL of 1 N HCl. The absorbance was measured at 450 nm (Molecular Devices, Spectramax 190, San Jose, CA, USA).

### 4.8. Statistical Analysis and Data Visualization

The experimental design for the plants was completely randomized, with five treatments including four STX IA transgenic lines (STX-5, STX-11, STX-12, and STX-13), a non-transgenic control, twelve replications, and one plant as the experimental unit. All data were tested for normal distribution and homogeneity at *p* ≤ 0.05, and then submitted to ANOVA, followed by the comparison of means using Tukey’s post-hoc test at *p* ≤ 0.05.

Fruit and juice quality parameters were assessed in a 2 × 5 factorial [2 treatments (non- or *C*Las-inoculated) × 4 STX IA transgenic lines + 1 control] arranged on a complete randomized design with five replications. All data were processed in R v. 4.0.2 (The R Foundation for Statistical Computing, Vienna, Austria) using the ExpDes, FactoMineR, and ggplot2 packages for the graphics and visualization of the statistical data.

## 5. Conclusions

Huanglongbing (HLB) is currently considered the most devastating disease for citrus production and has become a challenge for the scientific community, which has been pressured to develop novel strategies to manage this complex pathosystem. This disease has the potential to corner commercial citrus production. Therefore, the transgenic plants evaluated in this study may represent a promising strategy for managing HLB shortly, particularly with the endemic occurrence of HLB.

The genetic transformation and expression of the gene encoding the sarcotoxin IA (STX IA) peptide were not deleterious to the Pera sweet orange plants and fruit/juice. All STX IA transgenic lines expressed the *sarcotoxin IA* (*stx IA*) gene in plant tissues. The *Agrobacterium*-mediated transformation of the plant using the antimicrobial peptide (AMP) sarcotoxin IA does not induce immunity against ‘*Candidatus* Liberibacter asiaticus’ (*C*Las) but reduces its susceptibility by promoting a lower cumulative incidence of *C*Las-positive plants with moderate symptoms. The fruit and juice quality of the STX IA transgenic lines attends to the quality standards required by the citrus industry. Juice from the STX-11 and STX-12 fruit did not have any sarcotoxin IA protein residues revealed by immunoblotting, only juice from STX-5 and STX-13 fruit. On the other hand, the juice of all transgenic lines showed low traces of sarcotoxin IA peptide, as detected by the indirect ELISA test.

## Figures and Tables

**Figure 1 ijms-23-09300-f001:**
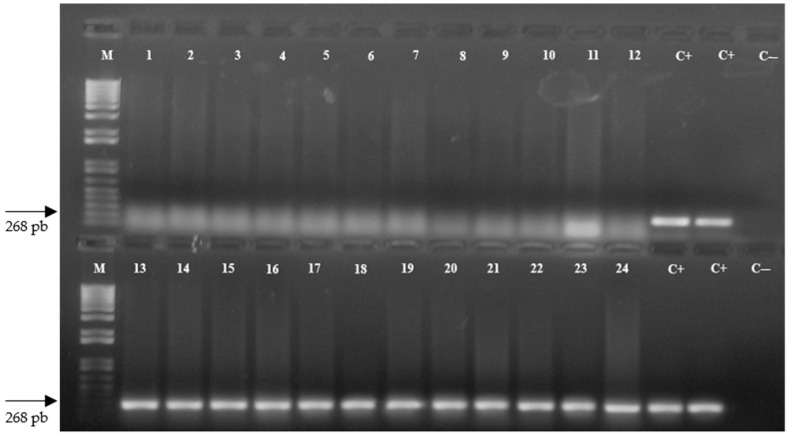
PCR amplification of the 268 bp DNA fragment of the *stx IA* gene integrated in the Pera sweet orange genome. Lanes: M, molecular marker; 1–12, non-transgenic Pera sweet orange plant (control); 13–15, plants of the STX-5 transgenic line; 16–18, plants of the STX-11 transgenic line; 19–21, plants of the STX-12 transgenic line; 22–24, plants of the STX-13 transgenic line; C+, positive control using the pST10 plasmid containing the *stx IA* gene; and C–, negative control.

**Figure 2 ijms-23-09300-f002:**
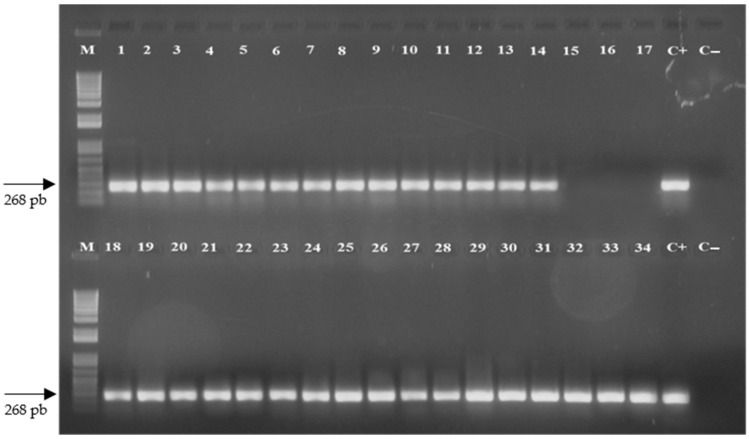
PCR amplification based on cDNA of Pera sweet orange plants to determine the expression of the *stx IA* gene. Lanes: M, molecular marker; 1–7, plants of the STX-5 transgenic line; 8–14, plants of the STX-11 transgenic line; 15–17, non-transgenic control plants (control); 18–26, plants of the STX-12 transgenic line; 27–34, plants of the STX-13 transgenic line; C+, positive control using the pST10 plasmid containing the *stx IA* gene; and C−, negative control.

**Figure 3 ijms-23-09300-f003:**
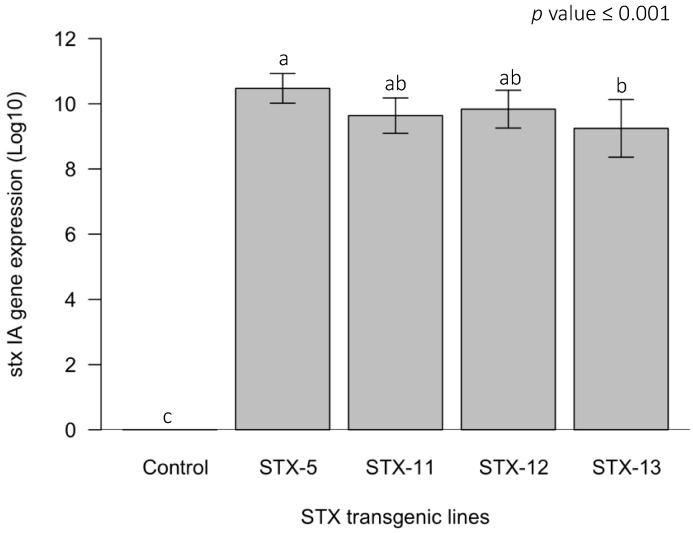
Quantification of the *stx IA* gene expression (Log_10_) by the Pera sweet orange transgenic lines STX-5, STX-11, STX-12, and STX-13, based on RT-qPCR test for the cDNA synthesized from extractions of total RNA. A non-transgenic Pera sweet orange plant was included as a negative control. Bars followed by the same letter do not differ according to the Tukey´s test (*p* ≤ 0.001).

**Figure 4 ijms-23-09300-f004:**
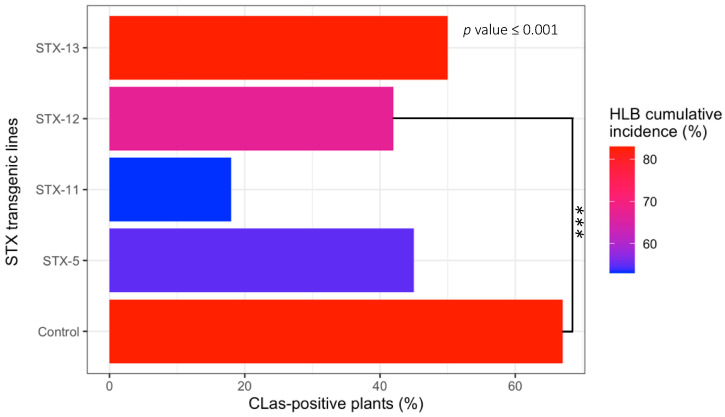
Cumulative incidence of huanglongbing (HLB, %) and ‘*Candidatus* Liberibacter asiaticus’ (*C*Las)-positive plants (%) of STX IA transgenic and non-transgenic Pera sweet orange plants. Significance level: ***, *p* ≤ 0.001.

**Figure 5 ijms-23-09300-f005:**
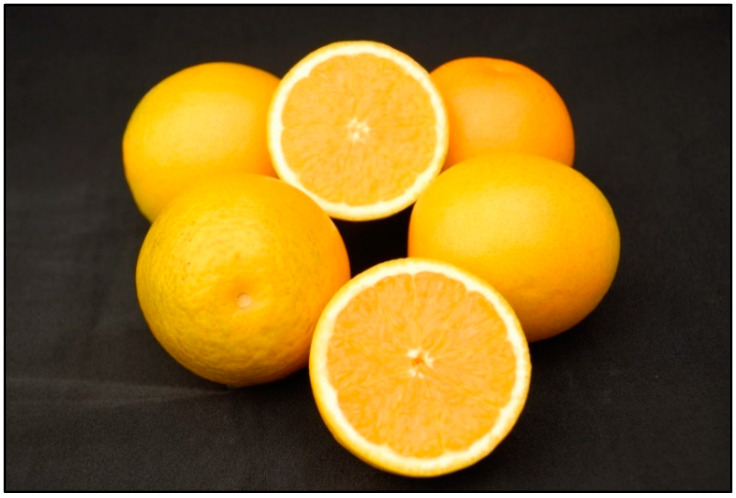
Fruit of ´*Candidatus* Liberibacter asiaticus’ non-inoculated (*C*Las^−^) STX-5 transgenic line of Pera sweet orange (*Citrus*
*× sinensis*) plants, produced under greenhouse conditions.

**Figure 6 ijms-23-09300-f006:**
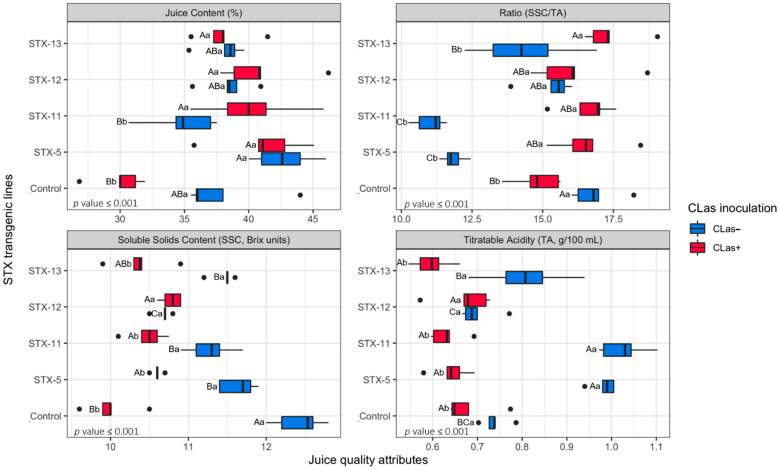
Juice (%) and soluble solids contents (SSC, °Brix), titratable acidity (TA, g·100 mL^−1^), and SSC/TA ratio for fruit from *Candidatus* Liberibacter asiaticus (*C*Las^+^)-inoculated and non-inoculated (*C*Las^−^) STX IA transgenic/non-transgenic control plants of Pera sweet orange (*Citrus*
*× sinensis*). Box plots followed by the same capital and lowercase letter do not significantly differ in regard to STX transgenic lines and *C*Las inoculation, respectively, according to the Tukey’s test. Significance level: *p* ≤ 0.001.

**Figure 7 ijms-23-09300-f007:**
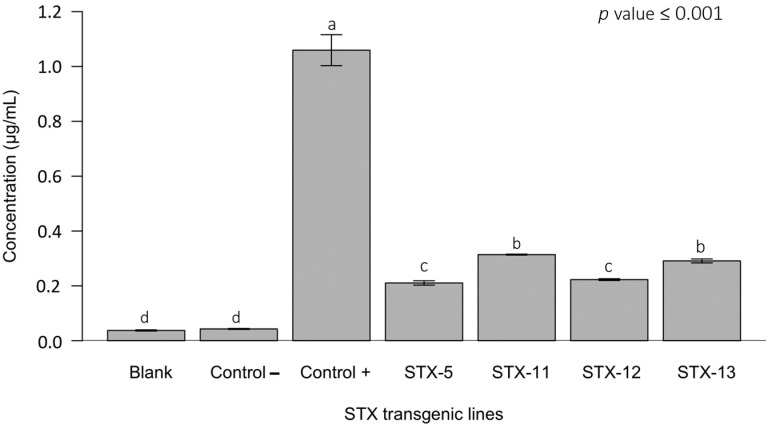
Concentration (μg·mL^−1^) of the sarcotoxin IA peptide in the juice of the STX IA transgenic and non-transgenic Pera sweet oranges by the indirect ELISA test. Blank, pre-immune serum; control −, non-transgenic negative control; control +, sarcotoxin IA synthetic peptide 1.0 μg mL^−1^; STX-5–STX-13, STX IA transgenic Pera sweet orange lines. Bars followed by the same letter do not differ according to the Tukey´s test (*p* ≤ 0.001).

**Figure 8 ijms-23-09300-f008:**
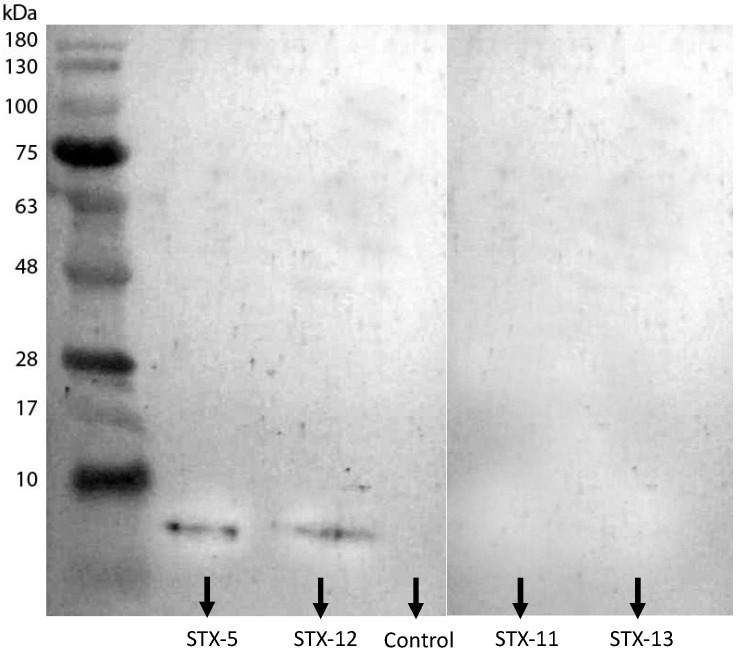
Accumulation of the sarcotoxin IA protein in the juice extracted from fruit of the STX IA transgenic lines STX-5, STX-11, STX-12, and STX-13, and non-transgenic control Pera sweet orange plants by Western blot analysis.

**Table 1 ijms-23-09300-t001:** Fruit quality of ‘*Candidatus* Liberibacter asiaticus’ (*C*Las^+^)-inoculated and non-inoculated (*C*Las^−^) STX IA transgenic and non-transgenic control plants of Pera sweet orange (*Citrus*
*× sinensis*).

**Source of Variance**	**Fruit Weight (g)**		**Fruit Length (mm)**		**Fruit Diameter (mm)**	
	***C*Las^+^**	***C*Las^−^**	***C*Las^+^**	***C*Las^−^**	***C*Las^+^**	***C*Las^−^**
Control	111 Bb ^1^	132 Ab	67.4 Bc	74.5 Aa	65.6 Bb	70.7 Aa
STX-5	151 Aa	138 Ba	72.3 Aa	68.6 Bb	71.8 Aa	66.0 Bb
STX-11	129 Aab	133 Ab	68.1 Bbc	71.7 Aab	68.8 Aa	70.0 Aa
STX-12	143 Aa	124 Bb	71.7 Aab	69.2 Ab	71.9 Aa	68.6 Bab
STX-13	131 Aab	135 Ab	69.7 Aac	70.6 Aab	69.0 Aa	70.2 Aa
CV (%)	9.11		3.21		2.54	
STX	***		ns		ns	
*C*Las	*		ns		ns	
STX × *C*Las	***		***		***	
	**Fruit Shape (L/D)**		**Normal Seeds ^2^**		**Aborted Seeds ^2^**	
	***C*Las^+^**	***C*Las^−^**	***C*Las^+^**	***C*Las^−^**	***C*Las^+^**	***C*Las^−^**
Control	1.03 Aa	1.05 Aab	4.5 Aa	1.6 Bab	4.0 Aa	3.3 Ab
STX-5	1.00 Ba	1.09 Aa	0.7 Bb	4.0 Aa	0.2 Bc	4.0 Aab
STX-11	0.98 Ba	1.02 Abc	0.2 Bb	1.0 Ab	0.1 Bc	3.4 Ab
STX-12	0.99 Aa	1.00 Ac	1.5 Bb	4.5 Aa	2.5 Bab	6.6 Aa
STX-13	1.00 Aa	1.00 Ac	5.7 Aa	4.3 Aa	1.4 Ab	1.8 Ab
CV (%)	2.42		37.68		25.79	
STX	***		***		***	
*C*Las	***		ns		***	
STX × *C*Las	**		***		***	

^1^ Means followed by the same capital letter in the row or lowercase letter in the column do not differ according to the Tukey´s test. ^2^ Data were transformed to arcsin $\sqrt{\left(x/100\right)}$ before submitted to ANOVA. Significance level: ns, non-significant; *, *p* ≤ 0.05; **, *p* ≤ 0.01; ***, *p* ≤ 0.001.

**Table 2 ijms-23-09300-t002:** Fruit and juice color attributes, including lightness (*L**), red/greenness (*a**), yellow/blueness (*b**), chroma (*C**), and citrus color index (CCI), of ‘*Candidatus* Liberibacter asiaticus’ (*C*Las^+^)-inoculated and non-inoculated (CLas^−^) STX IA transgenic and non-transgenic control plants of Pera sweet orange (*Citrus*
*× sinensis*).

**Source of Variance**	**Fruit Color**									
	***L****		***a****		***b****		***C****		**CCI**	
	***C*Las^+^**	***C*Las^−^**	***C*Las^+^**	***C*Las^−^**	***C*Las^+^**	***C*Las^−^**	***C*Las^+^**	***C*Las^−^**	***C*Las^+^**	***C*Las^−^**
Control	69.3 Bc ^1^	71.7 Aa	8.21 Aa	4.44 Bb	64.0 Ab	63.4 Ab	64.6 Ab	63.7 Ab	1.24 Bb	1.91 Ab
STX-5	73.9 Aa	72.5 Aa	7.41 Ba	10.3 Aa	72.8 Aa	72.2 Aa	72.2 Aa	73.0 Aa	1.37 Ba	2.08 Aab
STX-11	71.8 Aac	67.8 Bb	7.12 Aa	3.53 Bb	69.5 Aa	61.1 Bb	70.1 Aa	61.6 Bb	1.45 Aa	0.99 Ac
STX-12	72.1 Aab	71.8 Aa	9.40 Aa	10.8 Aa	72.6 Aa	69.0 Ba	73.4 Aa	70.0 Ba	1.77 Aa	2.19 Aa
STX-13	71.2 Bbc	73.4 Aa	7.31 Aa	4.79 Ab	68.7 Aa	69.0 Aa	69.2 Aab	70.3 Aa	1.57 Aa	1.08 Ac
CV (%)	2.10		28.09		3.70		3.73		31.80	
STX	***		***		***		***		*	
*C*Las	ns		ns		***		**		ns	
STX × *C*Las	***		***		**		**		*	
	**Juice Color**									
	***L****		***a****		***b****		***C****		**CCI**	
	***C*Las^+^**	***C*Las^−^**	***C*Las^+^**	***C*Las^−^**	***C*Las^+^**	***C*Las^−^**	***C*Las^+^**	***C*Las^−^**	***C*Las^+^**	***C*Las^−^**
Control	36.7 Bd	37.9 Ab	–3.04 Aa	–4.38 Bc	18.1 Bd	19.7 Ac	18.3 Be	20.2 Ac	–4.57 Ab	–5.85 Be
STX-5	38.0 Bb	38.6 Aa	–3.57 Ac	–4.20 Bb	18.4 Bd	21.3 Ab	18.8 Bd	21.7 Ab	–5.08 Ad	–5.09 Ac
STX-11	38.7 Aa	38.6 Aa	–3.87 Ad	–4.16 Bb	21.7 Ba	22.2 Aa	22.0 Ba	22.5 Aa	–4.60 Ab	–4.84 Bb
STX-12	37.8 Abc	37.0 Bc	–3.44 Bb	–2.99 Aa	18.9 Bc	22.3 Aa	19.2 Bc	22.5 Aa	–4.80 Bc	–3.62 Aa
STX-13	37.2 Bcd	37.9 Ab	–3.07 Aa	–4.55 Bd	19.5 Bb	21.0 Ab	19.8 Bb	21.4 Ab	–4.21 Aa	–5.71 Bd
CV (%)	0.62		1.25		0.86		0.83		1.15	
STX	***		***		***		***		***	
*C*Las	**		***		***		***		***	
STX × *C*Las	***		***		***		***		***	

^1^ Means followed by the same capital letter in the row or lowercase letter in the column, for each fruit and juice color parameters do not significantly differ according to the Tukey’s test. Significance level: ns, non-significant; *, *p* ≤ 0.05; **, *p* ≤ 0.01; ***, *p* ≤ 0.001.

**Table 3 ijms-23-09300-t003:** Sugar composition in juice extracted from fruit of ‘*Candidatus* Liberibacter asiaticus‘ (*C*Las^+^)-inoculated and non-inoculated (CLas^−^) STX IA transgenic and non-transgenic control plants of Pera sweet orange (*Citrus*
*× sinensis*).

Source of Variance	Sugars (g·L^–1^)						∑Sugars	
	Sucrose		Glucose		Fructose			
	*C*Las^+^	*C*Las^−^	*C*Las^+^	*C*Las^−^	*C*Las^+^	*C*Las^−^	*C*Las^+^	*C*Las^−^
Control	96.20 Bd ^1^	139.43 Aa	32.02 Be	53.78 Aa	40.16 Bd	54.66 Ab	168.39 Be	247.88 Aa
STX-5	91.39 Be	127.35 Ab	40.88 Bd	45.14 Ad	42.72 Bc	45.19 Ae	174.99 Bd	217.69 Ac
STX-11	117.69 Ab	114.71 Be	49.77 Ab	50.35 Ab	54.98 Aa	48.54 Bd	222.45 Ab	213.62 Bd
STX-12	125.76 Aa	122.31 Bc	52.82 Aa	47.56 Bc	55.47 Aa	55.52 Aa	234.06 Aa	225.40 Bb
STX-13	103.99 Bc	118.77 Ad	44.58 Bc	45.53 Ad	47.95 Bb	53.24 Ac	196.52 Bc	217.56 Ac
CV (%)	0.39		0.99		0.42		0.22	
STX	***		***		***		***	
*C*Las	***		***		***		***	
STX × *C*Las	***		***		***		***	

^1^ Means followed by the same capital letter in the row or lowercase letter in the column do not significantly differ according to the Tukey’s test. Significance level: ***, *p* ≤ 0.001.

**Table 4 ijms-23-09300-t004:** Organic acid composition in juice extracted from fruit of ‘*Candidatus* Liberibacter asiaticus’ (*C*Las^+^)-inoculated and non-inoculated (CLas^−^) STX IA transgenic and non-transgenic control plants of Pera sweet orange (*Citrus*
*× sinensis*).

Source of Variance	Organic Acids (g·L^–1^)								∑Organic Acids	
	Citric		Malic		Succinic		Ascorbic			
	*C*Las^+^	*C*Las^−^	*C*Las^+^	*C*Las^−^	*C*Las^+^	*C*Las^−^	*C*Las^+^	*C*Las^−^	*C*Las^+^	*C*Las^−^
Control	19.88 Bc	29.67 Ab ^1^	2.87 Ab	2.22 Ab	1.68 Ab	1.27 Bc	1.30 Ab	0.98 Bc	25.75 Bc	34.97 Ab
STX-5	20.26 Bc	23.76 Ad	5.32 Aa	3.68 Ba	2.10 Aa	1.47 Bbc	1.67 Ba	1.94 Aa	29.35 Bb	30.87 Ac
STX-11	21.33 Bb	31.24 Aa	6.34 Aa	2.81 Bab	1.92 Aab	1.99 Aa	1.69 Ba	1.92 Aa	31.29 Bab	37.97 Aa
STX-12	24.27 Aa	22.76 Be	5.64 Aa	3.42 Ba	0.91 Bc	1.79 Aab	1.45 ab	1.79 Aa	32.29 Aa	29.78 Bc
STX-13	19.02 Bd	28.02 Ac	6.09 Aa	2.68 Bab	0.47 Bd	1.90 Aa	1.61 Aa	1.49 Ab	27.20 Bc	34.10 Ab
CV (%)	0.80		10.64		10.50		7.20		2.67	
STX	***		***		***		***		***	
*C*Las	***		***		***		ns		***	
STX × *C*Las	***		***		***		***		***	

^1^ Means followed by the same capital letter in the row and lowercase letter in the column do not significantly differ according to the Tukey’s test. Significance level: ns, non-significant; ***, *p* ≤ 0.001.

## Data Availability

All data generated and analyzed during this study are presented in the published version and in the Appendix A of this article.

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
