# Peer review of "Transgenic Sweet Orange Expressing the Sarcotoxin IA Gene Produces High-Quality Fruit and Shows Tolerance to ‘Candidatus Liberibacter asiaticus’"

_ijms, 2022, doi:10.3390/ijms23169300_

Round 1

Reviewer 1 Report

Dear authors,

Manuscript ijms-1811850 entiteled "Plant Reaction and Fruit Evaluation of Transgenic Pera Sweet Orange Events Expressing the Sarcotoxin IA Gene under ‘Candidatus Liberibacter asiaticus’ Infection and authored by Talita Vigo Longhi , Deived Uilian de Carvalho, Izabela Moura Duin , Maria Aparecida da Cruz  and Rui Pereira Leite Junior targets a hot topic. The manuscript is potentially interesting for the journal readers and the scientific community as a whole. While I appreciated reading this manuscript and I impressed by the amount of work done (my self is actively working in the genetically engineered crops field) I have to mention that some issues needs authors attention before the papere meets journal standards and I could recomand it for acceptance. The major issues are listed below:

1. the title have to be changed ! it does not reflect the main findings of the manuscript.

2. Do you have any evidence that Anti-microbial peptides AMP sarcotoxin IA is active against Huanglongbing causal agent Candidatus Liberibacter asiaticus’ (CLas)? Please this issue have to be clearly discussed in the introduction section it will determine the fate of the manuscript ! without addressing this issue it will be difficult to recommand this paper for publication. The information you present "The majority of the insect AMPs are small and 83 cationic [43], showing effective activities against Gram-negative and occasionally Gram- 84 positive bacteria [38,41]" does not justify alone your strategy to manage HLB disease.

3. In material and methods section : have the amplified 268 bp DNA amplicons been sequenced to ascertain they belong to the the AMP sarcotoxin IA (stx IA) gene and encode AMP.

4. In discussion section please remove this sentence "These contradictory results may be associated with changes in the composition of the endophytic microflora of the citrus phloem due to the presence of the sarcotoxin IA peptide." or clearly indicate it is speculation or present your data about the endophytes composition difference between the presence and absence of sarcotoxin IA peptide.

5. reference 44 never mention that "These peptides have minimal toxic and allergic side effects to humans and plants, being a safe antibiotic for potentially use in medicine and agriculture" The authors clearly said in their discussion "These results suggest that the protein could be used as an antibacterial substance. But for that purpose, many more experiments must be done: first, it must be determined if the antibacterial protein is antigenic when injected into animals"

6. The presence of AMP sarcotoxin in juice should be indicated and clearly discussed with all the drawbacks that this could have. Please address this point because without addressing it the paper in misleading and could not be recommended for publications.

I encourage the authors to address all my comments and to provide a revised version that I could recommend for publications.

Best regards

Author Response

Dear. Editor and Reviewers,

We really appreciate your comments and suggestions for taking the time and effort to review our manuscript. The manuscript has been rechecked and the necessary changes have been made in accordance with the reviewers’ suggestions. The responses to all comments have been prepared and addressed herewith. All the revised text in the manuscript has been indicated using -color marker changes. We greatly appreciate the reviewers for the valuable and critical reviews.

Reviewer #1

Manuscript ijms-1811850 entiteled "Plant Reaction and Fruit Evaluation of Transgenic Pera Sweet Orange Events Expressing the Sarcotoxin IA Gene under ‘Candidatus Liberibacter asiaticus’ Infection and authored by Talita Vigo Longhi , Deived Uilian de Carvalho, Izabela Moura Duin , Maria Aparecida da Cruz  and Rui Pereira Leite Junior targets a hot topic. The manuscript is potentially interesting for the journal readers and the scientific community as a whole. While I appreciated reading this manuscript and I impressed by the amount of work done (my self is actively working in the genetically engineered crops field) I have to mention that some issues needs authors attention before the paper meets journal standards and I could recommend it for acceptance. The major issues are listed below:

Answer: Thank you very much for commenting on some very important issues. Your suggestions were actively accepted.

  1. the title have to be changed! it does not reflect the main findings of the manuscript.

Answer: The title was changed to better elucidate our findings: Transgenic Sweet Orange Expressing the Sarcotoxin IA Gene Shows Reduced Susceptibility to ‘Candidatus Liberibacter asi-aticus’, besides Producing High-Quality Fruit.

  1. Do you have any evidence that Anti-microbial peptides AMP sarcotoxin IA is active against Huanglongbing causal agent Candidatus Liberibacter asiaticus’ (CLas)? Please this issue have to be clearly discussed in the introduction section it will determine the fate of the manuscript! without addressing this issue it will be difficult to recommand this paper for publication. The information you present "The majority of the insect AMPs are small and 83 cationic [43], showing effective activities against Gram-negative and occasionally Gram- 84 positive bacteria [38,41]" does not justify alone your strategy to manage HLB disease.

Answer: Up to date, there is no available information in the literature regarding the effectiveness of the AMP sarcotoxin IA against CLas, which reinforces the novelty of our study.   

  1. In material and methods section: have the amplified 268 bp DNA amplicons been sequenced to ascertain they belong to the the AMP sarcotoxin IA (stx IA) gene and encode AMP.

Answer: The 268bp DNA amplicons were sequenced to assure that they belong to the stx IA gene.

  1. In discussion section please remove this sentence "These contradictory results may be associated with changes in the composition of the endophytic microflora of the citrus phloem due to the presence of the sarcotoxin IA peptide." or clearly indicate it is speculation or present your data about the endophytes composition difference between the presence and absence of sarcotoxin IA peptide.

Answer: This sentence was removed from our manuscript.

  1. reference 44 never mention that "These peptides have minimal toxic and allergic side effects to humans and plants, being a safe antibiotic for potentially use in medicine and agriculture" The authors clearly said in their discussion "These results suggest that the protein could be used as an antibacterial substance. But for that purpose, many more experiments must be done: first, it must be determined if the antibacterial protein is antigenic when injected into animals"

Answer: This sentence was removed from the discussion section.

  1. The presence of AMP sarcotoxin in juice should be indicated and clearly discussed with all the drawbacks that this could have. Please address this point because without addressing it the paper in misleading and could not be recommended for publications.

Answer: This issue was better discussed in the manuscript, we stated the importance of further research on the evaluation of AMPs in humans.

Reviewer 2 Report

The paper describes the generation and characterization of transgenic citrus plants overexpressing sarcotoxin IA gene. Some obtained plants showed some tolerance to HLB, but did not show immunity to the disease. However, given the importance of HLB for the citrus industry worldwide, the paper shows interesting new information. The authors performed a rather complete analysis of the transgenic lines and analysed the impact of transformation on fruit and juice quality. The paper is generally well written, but it needs some improvement.

line 21: please add the species from which the transgene was selected

line 99:what is cPCR?

line 149: Please call the transformed plants "transgenic plants" of "transgenic lines", not "transgenic event plants". please delete "events" throughout the manuscript.

lines 148-150 and lines 152-154: the authors mention "cumulative incidence" and "incidence" in the text and figures, and they measure it as percentage. It is not clear if "cumulative incidence" and "incidence" is the same parameter. Is this related to the number of plants with symptoms, or is it based on a score assigned to each replicate? This should be better specified in the methods

tables and figures: I would change "healthy" with "non-inoculated", and "HLB” with "inoculated" or "graft inoculated" in the whole manuscript. This could facilitate the readers to better understand the comparisons among treatments

Author Response

Dear. Editor and Reviewers,

We really appreciate your comments and suggestions for taking the time and effort to review our manuscript. The manuscript has been rechecked and the necessary changes have been made in accordance with the reviewers’ suggestions. The responses to all comments have been prepared and addressed herewith. All the revised text in the manuscript has been indicated using -color marker changes. We greatly appreciate the reviewers for the valuable and critical reviews.

Reviewer #2

The paper describes the generation and characterization of transgenic citrus plants overexpressing sarcotoxin IA gene. Some obtained plants showed some tolerance to HLB, but did not show immunity to the disease. However, given the importance of HLB for the citrus industry worldwide, the paper shows interesting new information. The authors performed a rather complete analysis of the transgenic lines and analysed the impact of transformation on fruit and juice quality. The paper is generally well written, but it needs some improvement.

line 21: please add the species from which the transgene was selected

Answer: The transgene was isolated from the flesh fly (Sarcophaga peregrina).

line 99: what is cPCR?

Answer: It is a conventional PCR, changes were made in the manuscript.

line 149: Please call the transformed plants "transgenic plants" of "transgenic lines", not "transgenic event plants". please delete "events" throughout the manuscript.

Answer: Thanks for these suggestion, transgenic events were modified to transgenic lines.

lines 148-150 and lines 152-154: the authors mention "cumulative incidence" and "incidence" in the text and figures, and they measure it as percentage. It is not clear if "cumulative incidence" and "incidence" is the same parameter. Is this related to the number of plants with symptoms, or is it based on a score assigned to each replicate? This should be better specified in the methods

Answer: It is related to the number of symptomatic plants over the evaluated period (2016–2021).

tables and figures: I would change "healthy" with "non-inoculated", and "HLB” with "inoculated" or "graft inoculated" in the whole manuscript. This could facilitate the readers to better understand the comparisons among treatments

Answer: We made the changes in the revised manuscript: inoculated or graft-inoculated (CLas+) and non-inoculated (CLas−).